# Policymakers' concerns linking tobacco and Indigenous communities in India: A qualitative analysis of parliamentary questions (1952–2022)

Shilpi Sikha Das[1,2☯], Upendra Bhojani[1,2☯]*, Nugehalli Srinivas Prashanth[1]

**1** Centre of Adivasi Health, Institute of Public Health, Bengaluru, Karnataka, India, **2** Centre for Commercial Determinants of Health, Institute of Public Health, Bengaluru, Karnataka, India

☯ Contributed equally

* upendra@iphindia.org

## Abstract

Tobacco use and associated disease burden remain high among Indigenous communities in India. Despite an overall decline in tobacco consumption over the last few decades, the social disparities in tobacco use have widened, with Indigenous communities experiencing the least decline. Existing tobacco control policies lack specific considerations for Indigenous communities. Hence, as part of a broader research initiative focusing on the health of Indigenous communities, we explored how parliamentarians in India have framed their concerns about tobacco in reference to Indigenous communities over time. We sourced digital transcripts of exchanges (parliamentarians asking questions and ministers responding to them) as part of the Question Hour sessions in the lower- and upper-houses of the Indian parliament between 1952 and 2022. We used thematic content analysis for 301 transcripts that linked tobacco and Indigenous communities. Overall representation of Indigenous communities in the Indian parliament remains very limited. The major concerns expressed by parliamentarians included (1) occupational health hazards and inadequate access to healthcare services faced by members from Indigenous communities working as bidi makers and tendu leaf collectors;(2) unfair wages and exploitative work conditions for these workers; and (3) perceived negative impact on tobacco-linked livelihoods resulting from trade and investment-related policies in the tobacco sector. Parliamentarians did not raise issues related to high tobacco use and tobacco-related harms among Indigenous communities in general, nor did they talk about the negative impact of tobacco on forests that are central to the lives of Indigenous communities. Public health research and advocacy efforts need to acknowledge the complex and multiple intertwining links between tobacco (industry) and Indigenous communities. There is a need to sensitise policymakers on the health and environmental impacts of tobacco while addressing the prevailing exploitation of workers from Indigenous communities in precarious tobacco supply chains and providing viable alternative livelihoods.

**Data availability statement:** Apart from the data contained in the manuscript, the transcripts of parliamentary questions (and answers) are available in public domain through web portals of the lower house (https://sansad.in/ls/questions/questions-and-answers) and the upper house (https://sansad.in/rs/questions/questions-and-answers) of Indian parliament.

**Funding:** This work was supported by the DBT Wellcome Trust India Alliance CRC grant (IA/CRC/20/1/600007 to PNS; IA/CRC/20/1/600007 to UB) and the DBT Wellcome Trust India Alliance (clinical and public health) Senior Fellowship (IA/CPH/22/1/506533 to UB). The funders had no role in study design, data collection and analysis, decision to publish, or preparation of the manuscript.

**Competing interests:** The authors have declared that no competing interests exist.

## Introduction

The widespread tobacco consumption in India poses a huge public health challenge. Tobacco consumption accounts for about 1.94 million adult deaths annually in India, apart from a significant morbidity burden [1]. In India, tobacco consumption varies widely across social groups. The report of the Expert Committee on Tribal Health set up by the Government of India highlighted that while tobacco use among Indigenous communities varied across states, in general, it remained high among Indigenous communities [2]. It indicated that, as per the National Nutrition Monitoring Bureau survey (2008–2009) covering nine Indian states, 35.9% of men and 6.3% of women in Indigenous communities (categorised as 'Scheduled Tribes' as per the Indian constitution) smoked tobacco [2]. Furthermore, 37.7% of men and 23.1% of women from Indigenous communities chewed tobacco [2]. The report indicated that the true prevalence of tobacco use among Indigenous communities might be higher, as the nationally representative survey (National Family Health Survey) in 2005–2006 indicated the overall prevalence of tobacco use among men from the Scheduled Tribe category to be 71.7%, much higher compared to men not belonging to the Scheduled Tribes (56.3%) [2]. A recent analysis of a nationally representative study (Longitudinal Ageing Study in India) also pointed high burden of tobacco use among Indigenous communities. It is estimated that in the year 2017–2018, the prevalence of tobacco use among Scheduled Tribes (aged 45 years and above) was 46% [3]. Another recent effort at meta-analysis of 39 studies involving 56,883 individuals from Indigenous communities across India estimated a pooled prevalence of tobacco use at 60% (66% among men, and 42% among women) [4]. This implies that public health policies/programs and the broader social determinants favourable to the reduction in tobacco use in India have not equitably benefited all the social groups.

This concern about a high level of tobacco use among Indigenous communities is not limited to India. Studies have found that in several countries, especially in Australia, New Zealand, the United States of America, and Canada, the prevalence of commercial tobacco use among Indigenous communities has been much higher compared to non-Indigenous communities [5]. The tobacco industry has been recognised as one of the important commercial determinants affecting the health and well-being of Indigenous communities [6]. Many countries have recognised this and have developed specific strategies to reduce such disparities [7]. For example, Australia, in its National Drug Strategy (2017–2026) identifies Aboriginal and Torres Strait Islander people as priority groups with a specific sub-strategy for these communities, indicating the need for careful, contextual, and participative interventions [8]. Canada's Tobacco Strategy prioritises Indigenous communities, ensuring specific funding to First Nations, Inuit, and Metis Nation people, enabling the development of their own plans to address commercial tobacco use [9].

India is home to 7305 ethnic groups that are officially notified as 'Scheduled Tribes'. According to the Census of India (2011), 104.5 million people belonged to the 'Scheduled Tribe' category, constituting 8.6 percent of the total population in India [10]. Indigenous communities in India are highly diverse in terms of culture, language, social organization, and habitat. In general, Indigenous communities have

low rates of literacy, access to health infrastructure, and poor nutrition and health indicators [10]. There are constitutional safeguards to protect and promote the rights of Scheduled Tribes, including reservations in higher education and election seats [11].

There have been several policy initiatives by the Government of India for reducing tobacco use [12]. and these initiatives seem to be making some positive impact [13,14]. However, there is no specific focus or consideration for Indigenous communities within these initiatives. As part of the Centre for Training, Research and Innovation in Tribal Health, a multi-institutional research initiative, we have been researching substance use (among other health issues) among Indigenous communities [15]. As part of this broader research agenda, we are conducting research on various aspects of substance use among Indigenous communities: from exploring sociocultural pathways of substance use among local communities to mapping policy discourse related to substance use among Indigenous communities over time. As parliamentarians play a crucial role in shaping policy discourse and policymaking, including for tobacco control, we aimed to study how parliamentarians in India framed their concerns related to tobacco and Indigenous communities in India over time. Our earlier analysis of parliamentarians' concerns related to tobacco in general - not specific to Indigenous communities - indicated that parliamentarians were increasingly concerned about health harms of tobacco and the need for tobacco control measures amidst continuing concerns about tobacco-linked revenues and livelihoods [16]. We anticipated that parliamentarians would be interested in issues affecting Indigenous communities, given that they are considered special groups who suffered historical injustice and who have constitutional safeguards [11]. In fact, there is a separate Ministry of Tribal Affairs within the Government of India [17]. Hence, we aimed to better understand how parliamentarians framed their concerns linking tobacco and Indigenous communities over time. Such understanding will inform how tobacco has been perceived and/or problematised concerning Indigenous communities in the Indian parliament, with implications for tobacco control advocacy efforts with policymakers in India.

## Methods

### Study design and setting

We used a qualitative approach wherein we analysed the digital transcripts of questions and answers that referred to tobacco and Indigenous communities in India as part of the parliamentary proceedings in both the houses of the Indian parliament. India is a union of states with a representative parliamentary democracy and a bicameral legislature. We chose to source the digital transcripts of questions and answers happening as part of the Question Hour of the lower (*Lok Sabha*) [18] and the upper (*Rajya Sabha*) [19] houses of the Indian parliament. The Question Hour is a platform wherein concerned ministers respond to the prior-submitted questions by any members of parliament. These questions can cover any aspect of government functioning and administration, as well as raising concerns of their electorates. Hence, the idea is to promote government transparency and accountability to the electorate [18,19]. Our study focused on analysing questions/answers starting from the year 1952 (the year Question Hour was mooted, and digital transcripts are available from the web portals of the parliament) till the end of the year 2022.

### Data collection

We searched the publicly available digital catalogues of parliamentary questions from both houses: Lok Sabha (Lower House) [20] and Rajya Sabha (Upper House) [21], separately using a combination of search terms 'tobacco' and 'tribal' within the full text of the transcripts without any search filters. We used synonyms/substitutes of these terms (for tobacco: *bidi, beedi, biri, tendu, gutka, cigarette, hookah, hukka, kharra, gul, mishri, mawa, gudakhu, nastaar, khaini, chillum,* cigar, *and cheroot*; for tribal: *adivasi, adiwasi, and* indigenous) in all possible combinations. A total of 2556 transcripts were sourced, with the oldest dating May 29, 1952, and the most recent dating July 26, 2022. Such a search strategy was arrived at after trying various searches to better understand how these search catalogues work (e.g., they do not recognise Boolean operators). The search results were exported into an MS Excel spreadsheet, and duplicates were removed

using the relevant function. The first author (SSD) read all the transcripts in full and further excluded those that did not contain at least one of the two categories of search terms (tobacco and tribal) or had these terms occurring incidentally (often in answers) while the transcript had no substantive relevance to the subject of our study. Fig 1 provides details about the sourcing and screening outcomes. The first author then discussed the shortlisted questions as well as potential exclusions that lacked clarity with the second author (UB) and made a decision about inclusion/exclusion based on consensus. In case of disagreement, the matter was referred to the third author (PNS), who discussed with the first two authors and made a final decision. We included a total of 301 transcripts for analysis in this study.

## Data management and analysis

The transcripts were typically a few pages long. An MS Excel spreadsheet was used to create a dataset wherein each row represented a unique question (and answer) asked in the parliament. Various attributes related to that question were recorded (e.g., date when asked, parliamentarian/s asking the question, the parliamentary house, the ministry that was asked that question). Some of the questions were asked by more than one parliamentarian, and some of the parliamentarians had asked more than one question related to tobacco and Indigenous communities. Hence, another MS Excel spreadsheet was used to create a dataset wherein each row represented a unique parliamentarian. Various attributes of

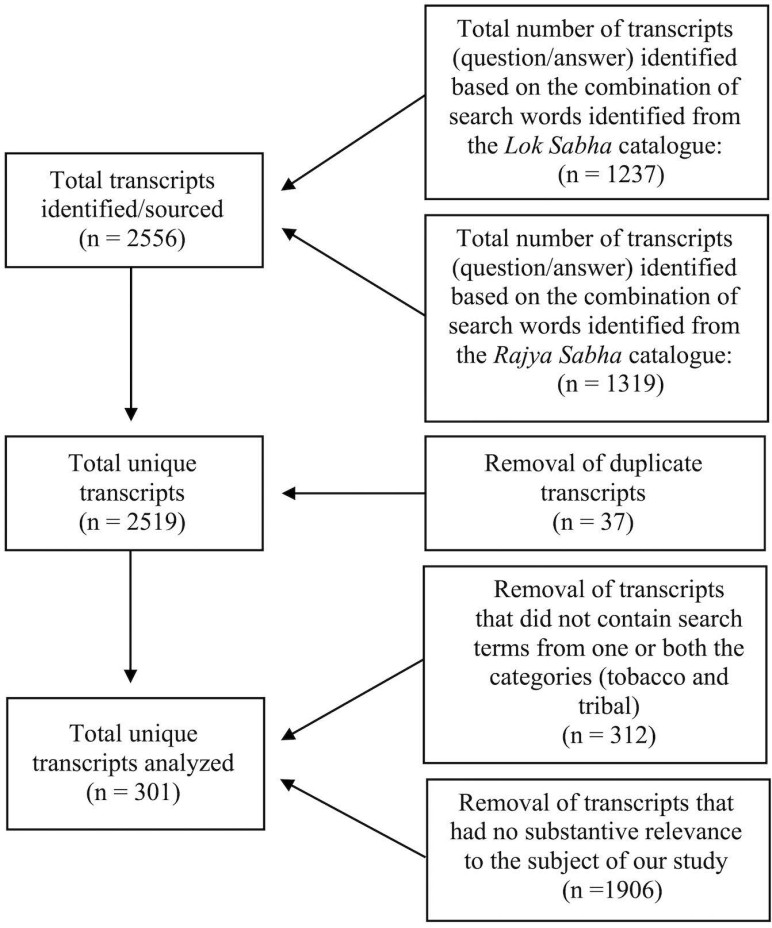

**Fig 1. Selection of transcripts of parliamentary questions for analysis (1952-2022).**

that parliamentarian were recorded (e.g., gender, political party affiliation, constituency, and the state that the parliamentarian represented).

We used thematic content analysis [22] using an interpretive lens to analyse transcripts. As this was an exploratory work with no precedent of similar inquiry (linking tobacco and Indigenous communities), the first author (SSD) used inductive coding of transcripts, guided by the overall question of 'how parliamentarians were framing their tobacco-related concerns in relation to the Indigenous communities?' rather than using any specific theoretical framework. She periodically discussed the coding pattern and the emerging themes she articulated based on coding with the second author (UB). Both these authors had formal training in qualitative research. The first author has worked with Indigenous communities for her doctoral and post-doctoral work [23,24]. The last author has been researching tobacco control for about two decades, including published work using parliamentary questions [16]. He has been collaborating with scholars working with Indigenous communities, including some of the members of the Indigenous community. Both authors, through periodic discussions and iterations, agreed on the overall themes and explored the interrelations across themes. These findings are presented using themes and specific quotes from the data. At some point, both datasets were combined through unique IDs given to transcripts to describe the data using attributes of the parliamentarians (presented in charts/tabular format).

## Results

### About the parliamentarians

A total of 257 parliamentarians asked 301 questions over the 70 years (1952–2022) linking tobacco and Indigenous communities. Most of them were in the lower house of parliament (76.7%) and were men (89.5%). Only 9.5% of these parliamentarians belonged to the Scheduled Tribe category. Over 60% of them belonged to the two major national political parties: Congress, accounting for 36.2% and Bharatiya Janata Party, accounting for 24.1% of these parliamentarians. The remaining 40% of parliamentarians belonged to 31 different political parties, including independent and nominated candidates. These parliamentarians represented constituencies across 28 states (including union territories and erstwhile jurisdictions), with Uttar Pradesh (15%), Andhra Pradesh (9%), and Maharashtra (8%) accounting for a third of these parliamentarians. Table 1 and Fig 2 provide details about select attributes of these parliamentarians. Nearly three-fourths (73.8%) of the total questions (analysed in this study) were asked in post 1990 period, which typically saw frequent tobacco control reforms. In general, questions spiked preceding major tobacco-related reforms (see Fig 3).

### Parliamentarians' concerns

We now describe parliamentarians' concerns about tobacco in relation to Indigenous communities in India using narratives and example questions under specific themes that emerged from analysis of the transcripts. Table 2 summarises these concerns by mentioning specific themes and issues within those themes.

### Health concerns

**The health of tobacco workers.** While many parliamentarians asked questions related to tobacco-related health harms, these concerns were about the population in general and not specifically about Indigenous communities. The reference to Indigenous communities, in such debates, was incidental as part of government responses where specific policies/programs related to Indigenous communities and/or forest areas were mentioned.

Whenever health-related concerns were brought up by parliamentarians that specifically linked tobacco and Indigenous communities, they were typically framed around Indigenous community members as workers in tobacco supply chains, especially tendu-leaf collectors, bidi rollers, and occasionally other aspects of tobacco supply chains (e.g., as tobacco barn owners or tobacco farmer/farm worker). Bidi is a smoked tobacco product, more prevalent than cigarettes in India, wherein processed tobacco is hand-rolled using tendu leaf (Diospyros melanoxylon) and tied with a cotton thread [25].

**Table 1. Parliamentarians asking questions about tobacco and Indigenous communities (1952-2022).**

| Political party | Total | Parliament House | | Gender | | Social Group | |
|---|---|---|---|---|---|---|---|
| | | *Lok Sabha (Lower House)* | *Rajya Sabha (Upper House)* | Men | Women | Scheduled Tribe | Others |
| Congress | 36.2% (93) | 36.5% (72) | 35% (21) | 37% (85) | 29.6% (8) | 28% (7) | 37.1% (86) |
| Bharatiya Janata Party | 24.1% (62) | 25.9% (51) | 18.3% (11) | 23% (53) | 33.3% (9) | 32% (8) | 23.3% (54) |
| Independent and nominated candidates | 3.9% (10) | 3% (6) | 6.7% (4) | 3.9% (9) | 3.7% (1) | 0% (0) | 4.3% (10) |
| All India Anna Dravida Munnetra Kazhagam | 3.5% (9) | 4.1% (8) | 1.7% (1) | 3.5% (8) | 3.7% (1) | 4% (1) | 3.4% (8) |
| Communist Party of India (Marxist) | 3.5% (9) | 3% (6) | 5% (3) | 3.5% (8) | 3.7% (1) | 4% (1) | 3.4% (8) |
| Telugu Desam Party | 3.1% (8) | 2% (4) | 6.7% (4) | 2.6% (6) | 7.4% (2) | 4% (1) | 3% (7) |
| Shiv Sena | 2.7% (7) | 3.6% (7) | 0% (0) | 2.6% (6) | 3.7% (1) | 0% (0) | 3% (7) |
| Biju Janata Dal | 2.3% (6) | 2% (4) | 3.3% (2) | 2.2% (5) | 3.7% (1) | 8% (2) | 1.7% (4) |
| Samajwadi party | 2.3% (6) | 3% (6) | 0% (0) | 2.2% (5) | 3.7% (1) | 0% (0) | 2.6% (6) |
| Dravida Munnetra Kazhagam | 1.9% (5) | 1% (2) | 5% (3) | 2.2% (5) | 0% (0) | 0% (0) | 2.2% (5) |
| Others (24 political parties combined, with each having fewer than 5 parliamentarians) | 16.3% (42) | 15.9% (31) | 18.3% (11) | 17.30% (40) | 7.5% (2) | 20% (5) | 16% (37) |
| Total | 100% (257) | 76.7% (197) | 23.3% (60) | 89.5% (23) | 10.5% (27) | 9.7% (25) | 90.3% (232) |

Some of the specific health concerns were related to inadequate nutrition and tuberculosis (with workers exposed to fine tobacco dust).

*"…whether the Government is aware that the workers engaged in beedi making are suffering from the occupational hazard of T.B. Tuberculosis is permanently there among bidi workers. Lakhs of workers are affected by this, but no medical care is provided to them…" (Oral Answers, Question 127, 1987)* [26]

The government cited experts' advice suggesting no factual basis to the widely perceived notion that tuberculosis is an occupational risk/hazard for bidi workers [26]. Nonetheless, these concerns continued over time.

*"…(a)whether there is high prevalence and mortality rate associated with tuberculosis among the women and bidi workers in the country…(d) the measures taken by the Government to control TB and funds spent therefor and the success achieved as a result thereof during the said period…" (Unstarred Question 2886, Lok Sabha, 2012)* [27]

The government responded by citing the Revised National Tuberculosis Control Program, indicating that the program data do not point to higher rates of mortality from tuberculosis among (women) bidi workers, and how the program universally promotes screening and access to free treatment for Tuberculosis to all, including women and bidi workers [27].

There were concerns about hunger and undernutrition among bidi workers from Indigenous communities.

*"…(a) whether Government's attention has been drawn to a new item captioned "Hunger stalks tribal beedi workers appearing in the Times of India" April 13, 1993? (b) if so, whether Government have obtained a report on the conditions of tribal beedi workers in Andhra Pradesh (c) what steps are proposed to prevent starvation amongst the section of labour? (Starred Question 11, Rajya Sabha, 1993)* [28]

The government responded by acknowledging that it has asked a report on this scenario from the state government of Andhra Pradesh [28].

 

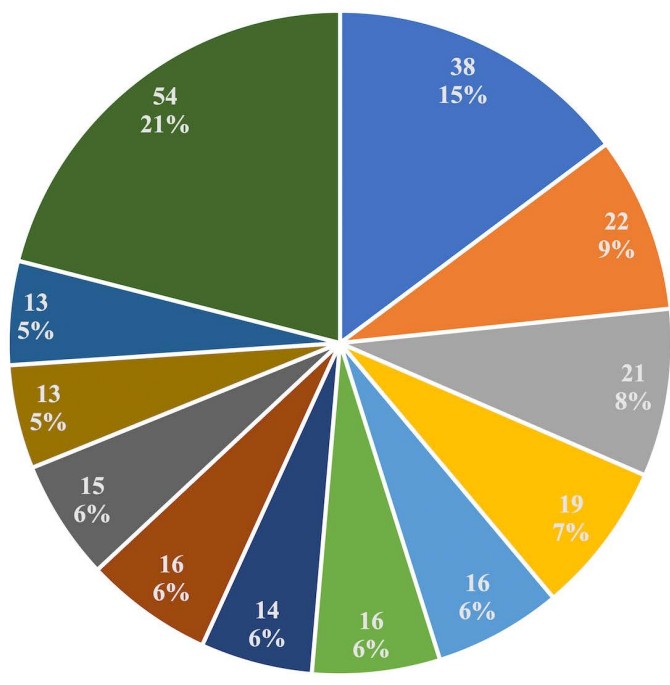

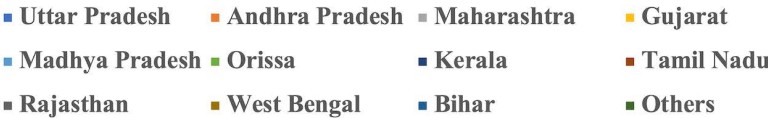

**Fig 2. Parliamentarians (state-wise) asking questions linking tobacco and Indigenous communities (1952-2022).**

**Health services for workers and their families.** Apart from specific health issues affecting workers, parliamentarians expressed concerns related to inadequate access to healthcare services or the functioning of publicly funded health insurance schemes for the workers and their families.

Given that there ought to be a specific policy providing physical healthcare delivery facilities for bidi workers, parliamentarians raised concerns about the presence and functioning of such facilities over time.

*"....(a) whether the Ministry has received a request for construction of hospital for Beedi workers at Gursahaiganj in Farrukhubad District; (b) if so, the decision taken thereon; (c) if no request has been received, whether Union Government propose suo moto to establish a hospital for these beedi workers in view of their long outstanding hardships on account of lack of basic medical facilities…"* (Starred-Supplementary Question 127, Lok Sabha, 1987) [26]

In this specific instance, the parliamentarian was also concerned about the delays in getting such a hospital in place despite repeated proposals. The government acknowledged the proposals for sanctioning a hospital and assured that it was under consideration while pointing out the expansion of dispensaries for bidi workers that was realised in the meantime [26].

Such voices demanding healthcare services and/or accountability for functioning health insurance schemes for the bidi and tendu workers persisted over time.

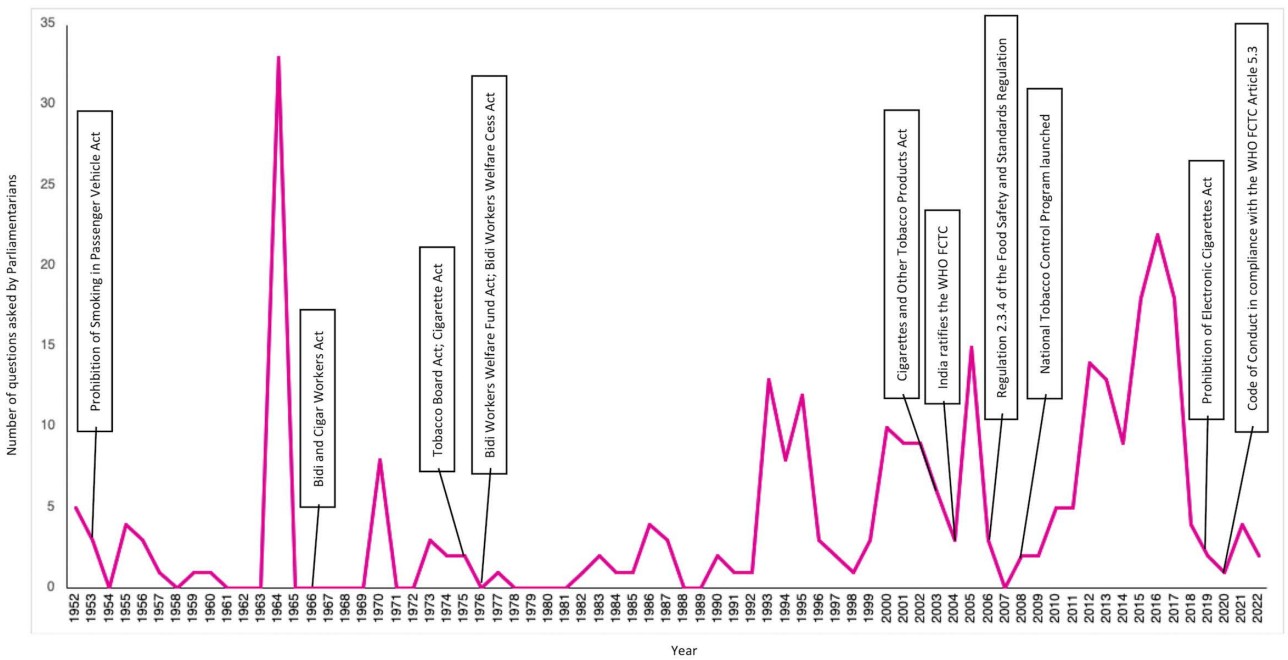

**Fig 3. Timeline of the questions in relation to the major tobacco-related policy reforms (1952-2022).**

**Table 2. Themes defining parliamentarians' concerns about tobacco and Indigenous communities in India.**

| Themes defining parliamentarians' concerns | Specific aspects/issues within each theme |
|---|---|
| Health concerns | Specific health issues affecting bidi workers<br>• Tuberculosis<br>• Hunger<br>Access to healthcare services by workers and their families<br>• Hospitals for bidi workers<br>• Basic healthcare services for *tendu* leaf collectors<br>• Coverage (workers and their families) and implementation of the publicly funded health insurance scheme (RSBY) |
| Labour welfare concerns | Bidi worker's welfare fund<br>• Contributions by employers (bidi industry) towards the bidi workers' welfare fund<br>• Effective allocation and utilisation of such funds by governments towards workers' welfare<br>Employees Provident Fund<br>• Facilitating enrolment of workers for the provident fund (social security)<br>• Irregularities by employers in depositing provident fund contributions<br>Identity cards for bidi workers<br>• Employers' role in recognising and issuing identity cards for workers<br>Contractualisation in the bidi industry<br>• Contract labour system in the bidi industry and exploitation of home-based bidi makers |
| Trade policy concerns | Collection and sale of *tendu* leaf<br>• Indigenous communities' ownership rights over *tendu* leaf collection<br>• Fair price to workers for the sale of tendu leaf<br>Nationalization of the tendu trade<br>• Potential impact on livelihoods and interests of tendu leaf pluckers<br>Foreign direct investment in the cigarette sector<br>• Potential impact on the bidi trade and hence workers within the bidi sector |

*"…(a) the number of tribal men and women in Orissa and Madhya Pradesh engaged to collect sal seeds, tendu leaves, Mahua and other forest and forests products…(c) whether any special programme for the health care of these tribal workers and their families have been formulated?" (Unstarred Question 4940, Rajya Sabha, 2000)* [29]

The government acknowledged that certain states (e.g., Andhra Pradesh and Madhya Pradesh) have come up with insurance for tendu leaf collectors and that Gujarat introduced a scheme to provide compensation to tendu leaf collectors for any injuries or deaths during working seasons [29].

*"…whether the Government is aware that RSBY centres in many districts are non-functional, if so, details thereof, State/ UT-wise, including Jharkhand along with the reasons therefor; and (e) whether the Government has taken note of irregularities in the above scheme being implemented in the interest of weavers in the country, if so, the details thereof and the action taken thereon along with the steps taken by the Government to increase its coverage particularly in tribal areas?" (Unstarred Question 4581, Lok Sabha, 2015)* [30]

The RSBY (*Rashtriya Swasthya Bima Yojana,* now integrated into the Pradhan Mantri Jan Arogya Yojana) referred to a national health insurance scheme initiated by the labour ministry (Government of India) that covered bidi workers as one of the 11 categories of workers, apart from covering those living below the poverty line. The government clarified that there are no central government-run RSBY Centres, as the scheme was being implemented by respective state governments [30]. The government further clarified that no specific irregularities were found regarding implementing RSBY for handloom workers [30].

## Labour welfare concerns

Many parliamentarians raised concerns about the realisation of labour rights and welfare measures for bidi workers, asking for the accountability of the government and employers in this regard.

**Bidi workers' welfare fund.** In 1976, the government of India enacted legislation that enabled the levy and collection of the cess on manufactured bidis towards creating a bidi workers' welfare fund [31] as well as using such fund towards the welfare of bidi workers [32] including the creation and improvements of public health and medical facilities, educational facilities for children of bidi workers, housing and recreational facilities, water supply and washing facilities and such other welfare measures.

Parliamentarians were concerned about the bidi manufacturers paying adequate duty (cess) towards the bidi worker welfare fund.

*"...(a) the amount collected as cess as on March 1995 from Bidi factories situated in tribal districts of Gujarat State; and (b) the amount incurred on the welfare for Beedi Workers during the above period?" (Unstarred question 6334. Lok Sabha, 1995)* [33]

*"…(e) whether the Government are aware of irregularities being committed by major industrialists to escape from cess on Beedi; and (f) if so, the efforts being made by the Government in this regard?" (Unstarred Question 3101, Lok Sabha, 1995)* [34]

There were concerns about how effectively such funds were allocated and utilised by governments towards workers' welfare.

*"...(a) whether the Government of Uttar Pradesh propose to allocate funds for the construction of houses of beedi workers particularly, in the tribal areas during the last three years." (Unstarred Question 1086, Lok Sabha 1998)* [35]

In these instances, the government assured to collect relevant information and disclose it to the upper house. The government assured that strict measures were being taken, including the surprise checks of bidi companies by the Central Board of Excise and Customs to prevent cess evasion.

**Provision of the employee provident fund.** Parliamentarians raised concerns about employers' role in enrolling and providing workers with the employee provident fund, a social security scheme allowing savings for the future, wherein workers ought to get benefits on retirement, resignation, or death, with a provision to also avail partial loans by workers for essential expenses. There were concerns that not all bidi workers were covered under the employees' provident fund.

*"…(a) the number of the Beedi workers covered under the Employees Provident Fund Scheme in Andhra Pradesh in the tribal districts, (b) whether the amount deducted from the salaries of the Beedi workers are being properly credited to the Provident Fund Accounts; (c) whether a large number of the Beedi workers have not been covered under the Provident Fund Scheme, and (d) if so, the steps taken by the Government to check the malpractice?.." (Unstarred question 6530, Lok Sabha 1995) [36]*

The government responded by providing the overall number (including but not specific to tribal districts) of bidi workers enrolled under the Employees Provident Fund (a total of 4,10,512 bidi workers as on 28 February 1995). The government assured that it was recovering the amount being deducted from workers' wages by employers and was crediting it to workers' provident fund accounts [36].

*"…(a) whether the government is aware of the complaints regarding fraudulent deposits in the provident funds of the beedi workers of Bihar and Gujarat due to which large-scale resentment is prevailing among them…" and further seeking information about any actions taken by the government in this regard. (Question 6332, Lok Sabha, 1995) [37]*

In this instance, the government responded that the relevant information is being collected and will be disclosed to the House.

**Identity cards for bidi workers.** One of the basic requirements for receiving such welfare benefits is to get recognised as a bidi worker by having an identity card issued by the employer to a bidi worker as statutorily mandated under the Bidi Workers Welfare Fund Act. There were concerns about many workers not getting the identity cards.

*"…(a) whether the Government have conducted any survey regarding the number of workers engaged in the Beedi Industry in certain States particularly in the tribal and backward areas of Gujarat State till September 30, 1995; (b) the number of identity card holders and non-identity card holders among them as on September, 1995, separately; (c) the efforts being made by the Government to liberate them from the harassment/oppression of contracts system…" (Unstarred Question 3101, Lok Sabha, 1995) [34]*

In this instance, the government assured to collect and disclose the information to the house.

**Contractualization in the bidi industry.** The contractors and sub-contractors started playing a major role for bidi companies, especially from the 1970s onward as bidi-making became largely a home-based work from what used to be a factory-based work. There remain concerns related to the exploitation of workers rolling bidis at home by the contractors.

*"…(c) the efforts being made by the Government to liberate them from the harassment/oppression of contracts system…" (Unstarred Question 3101, Lok Sabha, 1995) [34]*

The government acknowledged the concern and conceded that, given the historical developments resulting in the unique structure of the sector, it is "not feasible to abolish contract labour system" in the bidi sector [34]. However, it assured that

state governments have been requested to prevent exploitation of workers by ensuring better implementation of legal measures including, the beedi and cigar workers (conditions of employment) act 1966, the minimum wages act 1948, and formation of the tripartite committees (involving government, industry, and worker representatives) to address disputes and labour welfare issues [34].

### Trade policy concerns

There were many concerns expressed by parliamentarians on how the government policies related to domestic and international trade would shape the domestic bidi industry in general (which includes workers from Indigenous communities). These policies ranged from domestic decisions like taxation and nationalisation of tendu trade to measures like foreign direct investments in the tobacco sector. Here, we highlight concerns that specifically link trade-related policies with the livelihoods of workers from Indigenous communities engaged in bidi rolling and tendu leaf collection.

**Collection and sale of *tendu* leaf.** Tendu leaf, largely collected by Indigenous communities from the central Indian forests, is an important ingredient in the bidi industry. It is used to wrap bidi tobacco. With the tightening of forest conservation laws, parliamentarians raised concerns about access to these forests by Indigenous communities for collecting tendu leaves that formed an important forest-dependent livelihood option for them.

*"…(a) which minor forest produces would be purchased by Tribal Cooperative Marketing Development Federation to remove exploitation of the tribals by middlemen during the current financial year" (Unstarred Question 572, Rajya Sabha, 1990) [38]*

*"….(a) whether the Common Minimum Programme of the Government vowed to give ownership right to Minor Forest produce (MFP) especially Tendu leaves to tribals; (b) if so, whether many States have changed track to give ownership right of minor forest produce except Tendu leaves; (c) if so, the reasons therefor; (d) whether the Government has taken up the matter with the Ministry of Environment & Forests to convince States to grant ownership right to Tribals of Tendu leaves…" (Unstarred Question 3321, Lok Sabha 2004) [39]*

The government highlighted that while Indigenous communities residing in and around forests were already granted usufructuary rights (rights allowing persons to use a property not owned by them) for collecting minor forest produce for domestic consumption, there were no "clear-cut ownership rights" on minor forest produce. It suggested that while this matter belongs to the legislative business/mandate of state governments, the central government will request states to legislate, conferring such ownership rights to Indigenous communities for minor forest produce, including tendu leaves [39].

**Nationalisation of the *tendu* leaf trade.** In the 1960s and 1970s, many state governments nationalised the tendu leaf trade to enhance state revenues, prevent the exploitation of tendu leaf puckers, and ensure supply for small- and medium-sized industries. When a parliamentarian raised concerns about whether nationalisation would hurt Indigenous community members engaged in tendu leaf plucking, the government reiterated the protective impact of nationalisation on workers' rights.

*"...(a) whether Government are aware that the main profession of Adivasis in Uttar Pradesh in Mirzapur district is Tendu leaves; (b) whether Government are aware of numerous sufferings of these Adivasis due to nationalisation of tendu leave trade; and (c) whether Government propose to de-nationalise the tendu leave trade for the benefit of the weaker sections of society, i.e., Adivasis and if so. the salient features of the proposal?" (Unstarred Question 1008, Lok Sabha, 1977) [40]*

The government clarified that tendu leaf collection was a seasonal activity (typically from April to June) only partially contributing to the livelihoods of Indigenous communities. It further denied the parliamentarian's claim about its exploitative

impact and reiterated that the very intent of nationalising the tendu leaf trade has been to safeguard the interests of tendu leaf collectors, as they were not getting proper wages from contractors before such a move [40]. It clarified that the government was not considering any proposals to de-nationalise the trade [40].

Parliamentarians kept raising concerns about state governments not buying tendu leaves from Indigenous communities and/or not remunerating them adequately for such purchases.

*"….(a) whether millions of tribals in several districts of Madhya Pradesh are struggling for survival because of State Government refusal to purchase of tendu leaves from the tribals; (b) if so, the reasons therefore; and (c) the beneficiary oriented schemes introduced for providing alternate sources of incomes to the tribals displaced and deprived of collecting tendu leaves, sandalwood, etc." (Unstarred Question 1024, Lok Sabha 1992) [41]*

In this instance, the government assured to collect information on this from the respective state governments and disclose it to the house. It indicated that efforts are being made to promote local employment of Indigenous communities residing in and around forests as part of eco-development, nature conservation, and afforestation initiatives/schemes [41]. In response to other related questions by parliamentarians, the government indicated that tendu leaf was considered one of the important minor forest products to be promoted by the Tribal Co-operative Marketing Development Federation of India. [38].

**Foreign direct investment in the cigarette sector.** Parliamentarians also wondered if the promotion of foreign direct investment in the cigarette industry would negatively impact the domestic bidi industry and, hence, in turn, the manual labour within the bidi industry (which includes members from the Indigenous community).

*"…(a) whether the Government have reviewed its move to allow 100% Foreign Direct Investment (FDI) in cigarette industry; (b) if so, the major advantages of the economy by permitting such investment; (c) the extent to which it is likely to go against the interests of Indigenous bidi industry especially cottage industry and the workers engaged therein;…" (Starred Question 148, Lok Sabha,1999) [42]*

The government clarified that there was no ceiling in terms of foreign direct investment in cigarettes earlier. The government felt that cigarettes and bidis as different products and have very different market segments, and hence, enhanced foreign direct investments in the cigarette sector will only increase competitiveness in the cigarette sector without affecting the bidi industry or bidi workers [42].

Despite government assurances, parliamentarians expressed concerns about the potential exploitation of Indigenous community members in tobacco supply chains, demanding policy measures from the government to safeguard workers' interests. Such concerns were about tendu leaf collectors, bidi workers, as well as a minority of people who owned tobacco barns (facilities to cure tobacco leaves).

*"…(a) whether it is fact that ITDA in Andhra Pradesh has disbursed funds for tobacco barn construction in tribal areas of West and East Godavari districts in 1990, 1991 and 1992 (b) The details of funds distributed in the form of loans and in the form of subsidy to tribal tobacco farmers in the West Godavari district (c) Whether any monitoring has been done to ensure that the tobacco barns said to be owned by the tribals have not been taken over by the non-tribals and (d) in how many instances have been taken cases where such barns were constructed only with the view to draw subsidy?.." (Unstarred Question 4234, Rajya Sabha, 1994) [43]*

*"Whether Tendu leaf pluckers, most of whom are tribals in Madhya Pradesh, Chhattisgarh and Orissa, have demanded their share from the commercial exploitation of Tendu leaf and bamboos;…" (Unstarred Question 1886, Lok Sabha, 2002) [44]*

*"…(a) which minor forest produces would be purchased by Tribal Cooperative Marketing Development Federation to remove exploitation of the tribals by middlemen during the current financial year" (Unstarred Question 572, Rajya Sabha, 1990) [45]*

## Discussion

In this paper, through analysis of parliamentary questions over time (1952–2022), we describe major concerns that parliamentarians have expressed about tobacco in relation to Indigenous communities. We find that, overall, such concerns linking tobacco and Indigenous communities constitute a very small part (284 questions in the 70 years from 1952 to 2022) of the total tobacco-related questions by parliamentarians (1315 questions only in the 20 years from 1999 to 2019) [15]. Furthermore, a tiny proportion of parliamentarians expressing concerns linking tobacco and Indigenous communities came from Indigenous communities. Their concerns were largely about marginal workers from Indigenous communities in tobacco supply chains: about their health issues and access to healthcare services, their exploitation by employers, and potential threats to their interests and already precarious livelihoods. We now discuss some of these concerns by parliamentarians in the context of what we know from empirical studies on these issues, as well as their implications for tobacco control and public health measures among Indigenous communities.

### Invisible and precarious workforce within tobacco supply chains

Parliamentarians' concerns bring the spotlight on members of Indigenous communities that form a precarious workforce within tobacco supply chains (especially as bidi makers, tendu leaf collectors, and small-holder tobacco farmers), something that is hardly discussed in tobacco control discourse in India. Many studies corroborate concerns raised by parliamentarians about occupational risks and health hazards associated with such work, as well as economic exploitation and poor state of labour rights/welfare of these workers, despite the presence of specific regulatory provisions protecting these workers [46–48]. Studies highlight that members of Indigenous communities often take to these works in the absence of alternative livelihood opportunities, including a lack of, or poor implementation of, public schemes assuring employment guarantees [49,50].

While the WHO Framework Convention on Tobacco Control (a treaty signed and fully ratified by the Government of India) recognises the importance of promoting economically viable and healthy alternatives to tobacco-dependent livelihoods, there has been very limited progress on this front [51,52]. This study, along with corroborating evidence discussed earlier, implies the need for a just transition wherein the tobacco supply reduction measures shall go hand in hand with measures that ensure fair remuneration and work conditions for the marginal workers in tobacco supply chains and that promote access to economically viable and acceptable livelihood alternatives.

### Health concerns are largely limited to the 'worker identity'

What is intriguing is that parliamentarians' concerns about health issues, including access to healthcare services, are primarily about workers from Indigenous communities engaged in tobacco supply chains. We did not come across any concerns about tobacco use and its health harms among Indigenous communities in general, beyond workers in the tobacco sector. This is despite the fact that parliamentarians have been increasingly concerned about the health harms of tobacco in general, and tobacco use has remained disproportionately high among many Indigenous communities, who also experience poorer access to healthcare services/support. This could imply that parliamentarians are yet to recognise and prioritise the challenge of substance use (especially tobacco use in this case) in addressing the health of Indigenous communities. This could also reflect a poor representation of Indigenous communities in the Indian parliament. The fact that the Indigenous communities have experienced the least decline in tobacco use over the last few decades, it is

important to develop a special focus on Indigenous communities within tobacco control efforts, making them specifically relevant, acceptable, and informed by voices from Indigenous communities.

### Tobacco's impact on forests and the environment: a missing link?

Like the health harms of tobacco consumption, the negative impact of tobacco on forests and the environment seems to be another, more indirect, link to the health and well-being of Indigenous communities. The forests remain central to the survival and sustenance of forest-dwelling Indigenous communities. Tobacco, throughout its lifecycle, is known to harm the environment and forests [53]. In India, we have only started to map the magnitude of this problem: be it deforestation from tendu harvesting-related forest fires [54] and illegal logging of fuelwood for tobacco curing [55] or the environmental pollution resulting from tobacco production, consumption, and hazardous tobacco product waste [56].

Our findings and their discussion above have some implications for actions and further research related to tobacco control among Indigenous communities in India. Overall, it highlights the limited but diverse political concerns linking tobacco and Indigenous communities in India. We find that among the Indian Parliamentarians, but also in tobacco control discourse in general, there is very limited recognition of health consequences from the high overall tobacco use among many Indigenous communities, and the growing social disparity in tobacco use. It is desirable to develop a special focus on Indigenous communities within tobacco control efforts, making such efforts relevant, acceptable, and informed by voices from Indigenous communities. In this respect, we have seen a few examples in recent past including a webinar series and the first-of-its-kind national conference on tobacco use among tribal communities by the All India Institute of Medical Sciences (Deoghar) [57] as well as the efforts at understanding historical and socio-cultural pathways of substance use and creating context-specific tobacco-cessation support among Indigenous communities through the Centre for Training, Research and Innovations in Tribal Health [15]. However, we need to do more, including enhanced implementation, using context-specific strategies, of the national tobacco control program in districts with Indigenous communities.

The other important implication is to check the prevailing labour exploitation of Indigenous community workers, especially in bidi supply chains, while strengthening efforts in providing alternative (non-tobacco) livelihood options that are safer and viable. While there have been efforts to promote alternative crops in place of tobacco, such efforts have specifically targeted growers of cigarette tobacco [58] There do not seem to be major efforts from governments to promote alternatives for marginal workers in the bidi industry. Prioritising them under the prevailing wage employment promotion initiatives like the Mahatma Gandhi Rural Employment Guarantee Scheme [59] and the Pradhan Mantri Kaushal Vikas Yojana (Skill India Mission) [60] can help in this direction. While our study was limited to parliamentary concerns, future studies could look at other sources (such as news media, social movements, civil society voices) to map the broader public/policy discourse linking tobacco and Indigenous communities in India. It is time that the public health research and advocacy community expand its remit to engage with complex and multiple ways in which tobacco use and the tobacco industry are impacting Indigenous communities to make any equitable difference.

**Limitation.** There are limitations, mainly arising from the data, to our study. The digital search catalogues of the lower and the upper houses of the Indian parliament that we used to source the transcripts are well-optimised for specific searches, as they do not recognise Boolean operators. There is no specific guidance provided to optimise their use. This could have limited the number of relevant transcripts we could source despite the elaborate searches we ran. Apart from the low representation (in terms of absolute numbers) of the Indigenous communities in the Indian parliament, the study of the parliamentary proceedings over 30 years (1980–2009) by Ayyangar and Jacob [61] reveals that parliamentarians from scheduled tribe category, as well as women parliamentarians, participate less actively (asking a lesser number of questions on an average) in the Question Hour compared to men and parliamentarians from upper castes. This is an important limitation as the voices of parliamentarians from Indigenous communities remain limited, and so are the voices of women (most bidi makers and many of the tendu leaf collectors are women). They also highlight that the participation of parliamentarians from the north-eastern states of India remains limited – these are the states with a greater proportion of

Indigenous communities as well as a high prevalence of tobacco use. While we included all the question types, we did not include the debates – a long-form transaction specifically around legislative developments. While it is a limitation, it is also less likely that it would have added any substantive data, given that tobacco-related legislation does not have a specific emphasis on Indigenous communities.

The parliamentary questions are a limited component of the broader policy discourse. Furthermore, parliamentary questions are typically factual and concise, devoid of candid details. These questions are often framed with many considerations (like party agenda, political signalling, personal priority, individual interests) and do not always reflect the genuine concerns of constituents.

## Conclusions

Overall, the representation of Indigenous communities in the Indian parliament remains very limited. Parliamentarians, over time, expressed varied concerns linking tobacco and Indigenous communities in India. The major concerns included (1) occupational and other health hazards as well as inadequate access to healthcare services faced by members from Indigenous communities working as bidi makers and tendu leaf collectors; (2) unfair wages, exploitative work conditions and inadequate social security to these workers; and (3) perceived negative impact on tobacco-linked livelihoods resulting from trade and investment related policies in the tobacco sector. Parliamentarians did not raise issues related to high tobacco use and tobacco-related harms among Indigenous communities, as well as the negative impact of tobacco on forests that are central to the lives of Indigenous communities.

## Acknowledgments

We would like to acknowledge Soniya SM, pursuing her public health degree at the Karnataka State Rural Development and Panchayat Raj University, for her support in data management during her three-month internship with us.

## Author contributions

**Conceptualization:** Upendra Bhojani.

**Data curation:** Shilpi Sikha Das.

**Formal analysis:** Shilpi Sikha Das, Upendra Bhojani.

**Funding acquisition:** Upendra Bhojani, Nugehalli Srinivas Prashanth.

**Methodology:** Shilpi Sikha Das, Upendra Bhojani.

**Project administration:** Shilpi Sikha Das, Upendra Bhojani.

**Supervision:** Upendra Bhojani, Nugehalli Srinivas Prashanth.

**Validation:** Upendra Bhojani.

**Writing – original draft:** Shilpi Sikha Das.

**Writing – review & editing:** Shilpi Sikha Das, Upendra Bhojani, Nugehalli Srinivas Prashanth.

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
