## [Decision Letter · Decision Letter 0]

11 Aug 2025

PGPH-D-25-00953

Parliamentarians’ concerns about tobacco and Indigenous communities in India: a qualitative analysis of parliamentary questions (1952-2022)

Dear Author,

Thank you for submitting your manuscript to PLOS Global Public Health. After careful consideration, we feel that it has merit but does not fully meet PLOS Global Public Health’s publication criteria as it currently stands. Therefore, we invite you to submit a revised version of the manuscript that addresses the points raised during the review process.

EDITOR:

Dear Authors

Thank you for this manuscript.

Apart from few comments from learned reviewers, I have some additional points for you to consider.

1. rationale- Kindly clarify the title - Parliamentarians’ concerns about tobacco and Indigenous communities in India: a qualitative analysis of parliamentary questions (1952-2022)- is it tobacco use in indegenous communities.

2. Please consider improving your rationale. do you think parliament would see indegenous communities different from rest of the country while discussing tobacco. if yes, this can be added in the text. and also why this segregated reveiw particularly for indegnious communities would be useful and how overall results may vary and so our interpretation if whole population is considered.

line 99- please consider sharing a direct link to the data depository or cite it for ease of other readers.

table1: Please consider making table less complicated. I dont know how international audience would benefit from such table and data. maybe u can consider in power and in opposition or something like that. is this table also pointing towards TII. Less questions means more vested interest. - this is my veiw of the things- different people may think differently and paper then conveys a different meaning.

Discussion- Consider adding a para on the direct and indirect policy implications of the study.

and also on emerging recommendations frm the study.

Please ensure that your decision is justified on PLOS Global Public Health’s publication criteria  and not, for example, on novelty or perceived impact.

Journal Requirements:

Additional Editor Comments (if provided):

We look forward to receiving your revised manuscript.

Kind regards,

Madhur Verma

Academic Editor

Reviewers' comments:

Reviewer's Responses to Questions

**Comments to the Author**

1. Does this manuscript meet PLOS Global Public Health’s publication criteria?

Reviewer #1: Yes

Reviewer #2: Yes

2. Has the statistical analysis been performed appropriately and rigorously?

Reviewer #1: N/A

Reviewer #2: Yes

3. Have the authors made all data underlying the findings in their manuscript fully available (please refer to the Data Availability Statement at the start of the manuscript PDF file)?

Reviewer #1: No

Reviewer #2: Yes

4. Is the manuscript presented in an intelligible fashion and written in standard English?

Reviewer #1: Yes

Reviewer #2: Yes

Reviewer #1: Thank you for asking me to review this interesting manuscript, which analyses parliamentary transcripts concerning tobacco-related issues and indigenous communities in India. The paper is well-written and addresses a significant topic using a novel and valuable approach. However, there are several questions and concerns that need to be addressed (see below).

My overarching suggestion is that the paper should adopt a more outward-facing perspective. Specifically, it would benefit from linking its findings more clearly to the broader literature, both within India and internationally. While the study adds new knowledge, it does not yet fully situate its findings within existing gaps in the literature for a global audience.

In the introduction, it would be helpful for the authors to provide international readers with more context about indigenous communities in India. A general overview would support a better understanding of the scale and nature of the issues discussed, and allow for comparisons with indigenous populations in other parts of the world who often face similar challenges.

The background section could be strengthened by including information about any tobacco control laws or policies that specifically address indigenous populations. For example, are there any tobacco control campaigns or cessation services that target these communities? Even noting the absence of such initiatives would help to contextualise the study's findings.

The paper should also consider the issue of tobacco surveillance. Are indigenous populations included in surveys like GATS or GYTS? Are there any national surveys beyond the report already referenced that include relevant data?

Additionally, if available, information about tobacco industry marketing strategies targeting indigenous populations would be valuable in the background section.

Regarding the methods, it would be useful to clarify whether the transcripts analysed included parliamentary debates on specific tobacco control bills or related legislation. If such transcripts were omitted or unavailable, what impact does this have on the study? This could be addressed as a potential limitation.

It is also unclear how the inclusion or exclusion of transcripts was handled. Did two authors independently screen and select the transcripts, or did one author do so with subsequent review by the second? How were disagreements resolved? Depending on the process, this could be discussed as either a limitation or a strength.

As someone who is not a qualitative researcher, I found the description of the analytical approach unclear. Why did the authors not consider using framework analysis, which is known for its flexibility? While the research may be novel, parliamentary records have previously been analysed in the context of tobacco control. It is not evident whether the authors consulted prior studies of this kind. Providing more detail on the analytic method and referencing specific qualitative approaches would help clarify this.

The dataset spans a lengthy period of 70 years. However, the paper does not currently contextualise the data in relation to the timeline of tobacco control laws and policies in India. Temporal context matters: without knowing what else was happening when these issues were raised, the findings risk being presented in isolation. A timeline covering 1952–2022, showing when the analysed transcripts were produced and when key tobacco control laws were introduced, would be very helpful.

The discussion section does not adequately link the findings to existing literature, either from India or internationally. Indigenous communities around the world often experience high levels of tobacco use and related harms. It would be useful to know whether the findings of this study corroborate, expand upon, or contradict existing research.

The limitations section should focus on the limitations of the study design or methodology rather than the data itself. For instance, if the transcripts contain few voices of women or indigenous parliamentarians, that is an important finding in its own right—not a limitation.

Finally, the authors could strengthen the paper by proposing specific actions that should follow from their findings. What are the implications for policy? How should tobacco control laws or programmes be adapted in light of these insights? What recommendations can be made to parliamentarians or other stakeholders? The authors have identified important gaps—such as the neglect of indigenous health and environmental concerns—but they stop short of offering concrete suggestions for change.

Reviewer #2: Was alternate livelihood used as a key word?

In the discussion, it is mentioned that health concerns were not raised. But in results section there is discussion about tuberculosis. Please clarify. The sentence can be modified accordingly.

**Do you want your identity to be public for this peer review?** For information about this choice, including consent withdrawal, please see our Privacy Policy

Reviewer #1: No

Reviewer #2: **Yes:** Muralidhar M Kulkarni

---

## [Decision Letter · Decision Letter 1]

22 Dec 2025

PGPH-D-25-00953R1

Policymakers’ concerns linking tobacco and Indigenous communities in India: a qualitative analysis of parliamentary questions (1952-2022)

Dear Dr. Bhojani,

Thank you for submitting your manuscript to PLOS Global Public Health. After careful consideration, we feel that it has merit but does not fully meet PLOS Global Public Health’s publication criteria as it currently stands. Therefore, we invite you to submit a revised version of the manuscript that addresses the points raised during the review process.

We look forward to receiving your revised manuscript.

Kind regards,

Madhur Verma

Academic Editor

Journal Requirements:

Reviewers' comments:

Reviewer's Responses to Questions

**Comments to the Author**

Reviewer #1: All comments have been addressed

Reviewer #2: All comments have been addressed

Reviewer #3: (No Response)

publication criteria?

Reviewer #1: Yes

Reviewer #2: Yes

Reviewer #3: Yes

3. Has the statistical analysis been performed appropriately and rigorously?

Reviewer #1: N/A

Reviewer #2: Yes

Reviewer #3: Yes

4. Have the authors made all data underlying the findings in their manuscript fully available (please refer to the Data Availability Statement at the start of the manuscript PDF file)?

Reviewer #1: No

Reviewer #2: Yes

Reviewer #3: Yes

5. Is the manuscript presented in an intelligible fashion and written in standard English?

Reviewer #1: Yes

Reviewer #2: Yes

Reviewer #3: Yes

Reviewer #1: I am satisfied with the responses

Reviewer #2: All queries have been addressed

Reviewer #3: My comments have been attached in the PDF uploaded as comments. The authors have already revised their manuscript as per the previous peer review comments, which appear substantial and have improved the quality of the paper.

**Do you want your identity to be public for this peer review?** For information about this choice, including consent withdrawal, please see our Privacy Policy

Reviewer #1: No

Reviewer #2: **Yes:** Dr. Muralidhar M Kulkarni

Reviewer #3: No

---

## [Decision Letter · Decision Letter 2]

5 Feb 2026

Policymakers’ concerns linking tobacco and Indigenous communities in India: a qualitative analysis of parliamentary questions (1952-2022)

PGPH-D-25-00953R2

Dear Dr. Bhojani,

We are pleased to inform you that your manuscript 'Policymakers’ concerns linking tobacco and Indigenous communities in India: a qualitative analysis of parliamentary questions (1952-2022)' has been provisionally accepted for publication in PLOS Global Public Health.

Best regards,

Madhur Verma

Academic Editor

Thank you for making suggested changes.

Reviewer Comments (if any, and for reference):

Reviewer's Responses to Questions

**Comments to the Author**

Reviewer #1: All comments have been addressed

Reviewer #3: All comments have been addressed

publication criteria?

Reviewer #1: Yes

Reviewer #3: Yes

3. Has the statistical analysis been performed appropriately and rigorously?

Reviewer #1: N/A

Reviewer #3: Yes

4. Have the authors made all data underlying the findings in their manuscript fully available (please refer to the Data Availability Statement at the start of the manuscript PDF file)?

Reviewer #1: No

Reviewer #3: Yes

5. Is the manuscript presented in an intelligible fashion and written in standard English?

Reviewer #1: Yes

Reviewer #3: Yes

Reviewer #1: (No Response)

Reviewer #3: Dear Authors,

Thanks for addressing the comments. I must say this is a wonderful work, a bit out of box thinking in the sense that it indirectly emphasizes how policymakers look into tobacco control in vulnerable population (here the tribal & indigenous communities) in India. This paper, if published, can guide changes in policymaking w.r.t. tobacco control in this country with a focus on the indigenous communities. The manuscript should be published.

Thanks

Santanu

**Do you want your identity to be public for this peer review?** For information about this choice, including consent withdrawal, please see our Privacy Policy

Reviewer #1: No

Reviewer #3: **Yes:** Santanu Nath
